# Effect of Blend Composition on Barrier Properties of Insulating Mats Produced from Local Wool and Waste Bast Fibres

**DOI:** 10.3390/ma16010459

**Published:** 2023-01-03

**Authors:** Anna Kicińska-Jakubowska, Jan Broda, Małgorzata Zimniewska, Marcin Bączek, Jerzy Mańkowski

**Affiliations:** 1Department of Innovative Textile Technologies, Institute of Natural Fibres and Medicinal Plants-National Research Institute, Wojska Polskiego 71b, 60-630 Poznan, Poland; 2Faculty of Materials, Civil and Environmental Engineering, University of Bielsko-Biala, Willowa 2, 43-309 Bielsko-Biala, Poland

**Keywords:** fibre waste, wool, flax, hemp, nonwoven, sound absorption, noise reduction, thermal resistance, thermal conductivity

## Abstract

This paper concerns the management of natural waste fibres. The aim of this research was the production of multifunctional acoustic and thermal insulation materials from natural protein and lignocellulosic fibre wastes, according to a circular bioeconomy. For the manufacture of the materials, local mountain sheep wool and a mixture of bast fibre waste generated by string production were used. Insulating materials in the form of mats produced by the needle-punching technique with different fibre contents were obtained. The basic parameters of the mats, i.e., the thickness, surface weight and air permeability were determined. To assess barrier properties, sound absorption and noise reduction coefficients, as well as thermal resistance and thermal conductivity, were measured. It was shown that the mats exhibit barrier properties in terms of thermal and acoustic insulation related to the composition of the mat. It was found that mats with a higher content of the bast fibres possess a greater ability to absorb sounds, while mats with higher wool contents exhibit better thermal insulation properties. The produced mats can serve as a good alternative to commonly used acoustic and thermal insulating materials. The production of the described materials allows for a reduction in the amount of natural fibre waste and achieves the goal of “zero waste” according to the European Green Deal strategy.

## 1. Introduction

Fibres of natural origin, including sheep wool, linen and hemp, are materials with high potential to be used in the manufacture of products with barrier properties in the field of acoustic and thermal insulation. Recently, there has been an increase in interest in ecological insulation materials made from renewable raw materials as well as waste materials, including textile waste [1,2,3,4,5,6,7].

The entire value chain of natural textile production based on raw materials such as sheep wool or fibres of plant origin from the cultivation of fibre crops and sheep farming, from processing to final bio-products, fits into the goals of the European Green Deal strategy, which indicates the need to take specific actions to achieve climate neutrality by 2050 and prevent environmental decline. One of the important goals is the pursuit of eliminating plastic goods from everyday life by replacing them with renewable, natural products and using recycled raw materials in their manufacture [8,9,10,11,12]. In the current situation, in the face of serious threats to the climate and the environment, it is extremely important to focus on the effective use of native textile natural raw materials, such as sheep wool, which is increasingly treated as a by-product of animal husbandry intended for meat. Another natural raw material is waste bast fibres produced during various processes. The use of these raw materials can be combined for new applications [13].

Materials based on natural fibres are gaining importance due to their environmentally friendly nature. Natural fibres are harmless to human health as they do not contain toxic substances [14,15]. Fibres of animal and plant origin, as renewable, fully biodegradable raw materials, meet the requirements of, for example, green construction. The use of green materials in construction is considered a potential way to minimise its negative effects on the natural environment [16]. In the construction industry, increasing attention is being paid to barrier materials for acoustic waves [17]. Building acoustics is understood as a set of parameters characterizing the transmission of sound through building partitions, e.g., partition walls, ceilings, facades and windows, while room acoustics refers to sound propagation in the closed space of rooms. Sound reduction in rooms is achieved through the use of products characterised by a high ability to absorb sounds and good acoustic insulation. The sound absorption capacity of a material is typically represented by the reverberation sound absorption coefficient, α, as a function of frequency. The noise reduction coefficient (NRC) is also used to characterise the acoustic barrier of the product. [18,19,20,21,22,23,24]. According to reports in the literature [3,25,26,27], the noise reduction coefficient for bast fibres and wool varies in range of 0.27 to 0.69 and depends on the type of raw material and sample thickness.

The effectiveness of sound absorption is determined by the shape of the room and the materials and objects used for its finishing [28,29]. One of the groups of products offered on the market in the field of acoustic insulation are various types of mats or panels intended for insulation of roofs, walls or ceilings. Such products are used mainly for the insulation of confined spaces. A second group are products used to suppress noise inside rooms, such as inside houses, offices and public utility buildings. Such products include carpets, rugs, curtains, felts and various types of nonwoven products. Products of this type absorb sound well in the medium and high frequency range, but poorly in the low frequency range. Other types of acoustic insulation materials are screens/partitions and wall/ceiling panels. Screens and partitions are often used in offices to improve the acoustics of the workplace and create favourable working conditions for people in open office arrangements. Soundproofing panels placed on walls and ceilings are also a form of decoration, improving both the acoustic comfort and the aesthetic value of the space. Currently, a large portion of products used as sound insulation are made from mineral wool and glass wool, as well as synthetic raw materials, such as polyurethane (PUR) foams.

In recent years, natural raw materials have become an important alternative to commonly used insulation materials due to their lower production costs and environmentally friendly nature. In the available literature, one can find reports that sheep wool is a good sound-absorbing material in the medium and high frequency range, with sound absorption values comparable to mineral wool and polyurethane foams [30,31]. Wool-based insulation materials are usually produced in the form of nonwoven products, the structure and parameters of which affect the acoustic barrier parameters [32]. Nonwovens have a porous structure with numerous spaces between the fibres. Porous materials tend to absorb more sound than smooth materials due to the open pore channels through which sound waves can penetrate. The acoustic properties of non-woven mats depend on many factors, such as the type and structure of fibres and parameters of the produced nonwovens, e.g., thickness, density and air permeability. It has been shown that the value of the sound absorption coefficient increases with increasing material thickness; however, at high frequencies of acoustic waves, the thickness of the nonwoven fabric layers has little effect on the sound absorption through the multilayer structure [33]. Researchers [34] have shown that coarse wool from local coloured mountain sheep, often treated as waste and a by-product of sheep farming, can be used to produce sound-absorbing materials in the form of felt or fabrics. The sound absorption of a felt made from randomly oriented, tangled and physically bound fibres mainly depends on the packing density of the fibres and the thickness of the felt. 

Reported research [35] has confirmed that flax fibre, tested in terms of sound absorption in the form of a loose mass of fibres with a thickness of 50 mm, also shows very good sound absorption properties in the high frequency range. In addition, researchers have shown that hemp fibres are the ideal alternatives to conventional sound absorbers [36,37,38]. 

It is worth mentioning that sheep wool, thanks to its good thermal properties, is used in construction as a heat insulator [39]. Products based on wool in the form of mats or a loose mass with a fibre backing can be used to insulate confined spaces, e.g., roofs, walls and ceilings. Commonly used insulating materials in the above-mentioned configurations are polystyrene and rock wool. The main disadvantages of these materials result from the method of their production, in which non-renewable raw materials are used. They are not biodegradable and their production processes have a negative impact on the natural environment. These materials show limited water vapor permeability [40]. Wool is a renewable, biodegradable and flame-retardant material and also has the ability to regulate the relative air humidity in a room. Wool fibres are characterised by high hygroscopicity, in contrast to products made from glass fibre, and they can absorb and desorb moisture without reducing thermal properties, which means that the barrier properties of wool remain at a similar level under changing air humidity conditions. Due to their chemical composition, wool fibres can absorb more than 35% of moisture in relation to their weight without feeling wet. Wool also contributes to maintaining a constant relative air humidity in the room [39].

Hegyi A. et al. [40] investigated sheep wool mats in which an additional 10–15% of thermoplastic fibres were added to the wool fibre mass. The mats were made by thermal welding. The developed wool thermal insulation materials with improved mechanical properties have a positive effect on the health of users, thanks to their good water vapor permeability and the ability to store/release moisture from the environment, which in turn allows the regulation of air humidity in the room and eliminates the risk of condensation and mould formation.

The fact that bast fibre contains a channel along its axis makes it suitable for use as thermal insulation [41]. The interest in using hemp in building materials is closely related to its properties such as high hygroscopicity, good mechanical properties and air permeability, which translate to the products made from them. Materials based on bast fibres, through their ability to regulate air humidity in a certain range, contribute to the creation of a microclimate that is beneficial to inhabitants. Such technological solutions are safe for users and are environmentally friendly. Pennacchio et al. [42] studied sheep wool in combination with hemp fibres in the form of semi-rigid panels for thermal and acoustic insulation. The panels were made from chopped hemp fibres and poor-quality wool that was unsuitable for textile processing, i.e., it had coarse and irregular length fibres. The fibres were glutted thank to treatment with sodium carbonate solution. The developed 45 mm-thick semi-rigid panels showed a thermal conductivity coefficient of 0.041 W/(m·K), tested at 25 °C.

The available literature does not contain reports into using waste natural proteins and lignocellulosic fibres for the production of bifunctional insulation materials. Hence, the aim of the current study is to develop mats made from a mixture of wool and bast fibres with barrier properties in terms of thermal and acoustic insulation. The study was conducted within a PhD Thesis titled: The influence of the composition of the wool fibre blend from sheep from Polish mountain areas and bast fibres on the barrier properties of mats intended for thermal and acoustic insulation [13]. This research attempts to effectively use domestic natural fibrous resources in the form of wool from mountain sheep breeds, which is often treated as waste, and waste flax and hemp fibres produced at the carding stage during twine production. Bearing in mind the “zero waste” strategy, using low-quality raw materials and semi-finished products, which often constitute unused waste, in the production of barrier materials for acoustic and thermal insulation presents an environmentally friendly alternative for their use. Insulating nonwovens with different percentages of sheep wool and bast fibres were made by needle-punch technology. The acoustic and thermal properties of the manufactured materials were analysed and compared with other products.

## 2. Materials and Methods

### 2.1. Materials

The research material consisted of mats made from washed sheep wool from Polish mountain sheep from the Centre for Regional Products in Koniaków and waste bast fibres produced during the two-stage carding process in the production of flax and hemp twine. The mats were produced at the Research Plant “Lenkon” at the Institute of Natural Fibres and Medicinal Plants in Stęszew with the use of an industrial technological line for the processing of flax and hemp fibres.

#### Manufacture of Mats

Mats were manufactured using three components: (1) a mixture of 75% hemp and 25% flax fibres, (2) sheep wool and (3) jute net. The wool obtained from sheep was in the form of an inner and outer cover. The wool used for the production of mats was characterised by a fibre diameter of 77 µm (SD = 7 µm) for the outer cover and a fibre diameter 41 µm (SD = 12 µm) for the inner cover, and the average fibre length was 108 mm (SD = 30 mm). The mixture of flax and hemp waste fibres was characterised by a fibre length of 69 mm and the impurities content was 18%. For technological reasons, the production of mats required the use of a jute net to strengthen the structure of the non-woven fabric. For each mat composition, the applied jute net was the same. The jute net was characterised by a mass per square meter of 69 g/m^2^ (SD = 3.8 g/m^2^), a thickness of 1.1 mm (SD = 0.04 mm). The number of warps was 13 threads/10 cm (SD = 0.4) and the number of wefts was 10 threads/10 cm (SD = 0.4). 

The mats were produced using an industrial technological line including carding machines, fleece forming parts and a JM-1800M needle-punching machine, adapted to work with flax and hemp fibres. The same parameters of the mat forming process were used for all raw material variants: the number of layers was 18, the speed of the fleece backing was 0.8–1.0 m/min and the needling density was 50/cm^2^. For the production of 100% wool (W) and 100% bast (B) one-component mats, 40 kg of each raw material was used. The following fibre ratios were used for the production of mixed mats:-10 kg wool and 30 kg bast fibres (mat composition 25% W/75% B);-20 kg wool and 20 kg bast fibres (mat composition 50% W/50% B);-30 kg wool and 10 kg bast fibres (mat composition 75% W/25% B).

The mat forming machine was manually fed. At the carding stage, the fibres were loosened and mixed. In the next stage, a thin fleece layer was formed. The fleece layers were staked one on top of the other and placed on the jute net and then transported on the jute net to the next sections of the device. In the following steps, the fleece layers were needle-punched. The final stage of mat production involved rolling the manufactured mat onto a roller. The produced mats are shown in Figure 1.

### 2.2. Methods

The following parameters were determined for the mats:Thickness in accordance with PN-EN 9073-2. The test was performed using an electronic thickness gauge J-40-V (Checkline, Germany), with a load of 0.5 kPa.Surface density in accordance with PN-EN 29073-1. The tests were performed using a Mettler PM 480 electronic balance (Mettler Toledo, Switzerland).Air permeability in accordance with PN-EN ISO 9237. The tests were carried out using the Air Permeability Tester III FX 3300 apparatus (Textest Instruments Switzerland), at a pressure of 200 kPa.Sound absorption coefficient (α) in accordance with PN-EN ISO 10534-1. The tests were performed using the standing wave method with the use of an impedance tube by Brüel & Kjaer 4002 containing one microphone. Measurements were made for round samples with a diameter of 100 mm at third- octave intervals in the frequency range of 250–3150 Hz. During the measurements, a one-dimensional acoustic field was generated inside the tube. The plane sound wave was propagated in the tube and then reflected at the other end of it. As a result of the interference of the reflected wave with the incident wave inside the tube, a standing wave was formed with sound pressure maxima and minima. The sound absorption coefficient was calculated based on the ratio of the measured maximum and minimum standing wave sound pressures.Noise reduction coefficient (NRC) was calculated as the average value of the sound absorption coefficients at the frequencies of 250, 500, 1000 and 2000 Hz.Thermal resistance (R_ct_) in accordance with PN-EN ISO 11092. The tests were carried out using a heat-insulated SGHP-8.2 sweating plate with a CEO 910-4 environmental chamber (Measurement Technology, Whitestown, IN, USA).Thermal conductivity coefficient (λ) in accordance with PN-EN 12667. The test was performed with the use of the LaserComp FOX314 apparatus (TA Instruments, New Castle, DE, USA). The tests were carried out with the use of a fixed, constant flow of heat flux passing through the tested samples. The temperature range used was from −20 °C to + 55 °C with a temperature difference of 25 °C.The statistical evaluation of the test results was performed on the basis of the non-parametric ANOVA Kruskal–Wallis test at a significance level of 0.05 using the Statistica software 8.0.

## 3. Results

### 3.1. Properties of Materials

Table 1 presents the results of metrological tests of the mats.

The thicknesses of the tested mats were in the range of 6.8–10.6 mm. The 100% B mat was the thinnest. In the case of the mixed mats, the thicknesses of the mats were higher and reached values over 10 mm, while the 100% W mat was 8.6 mm thick. The surface densities of the mats made of bast and wool fibres were in the range of 986 to 1491 g/m^2^. The 100% W and 50% W/50% B mats were characterised by the lowest surface density. The values of air permeability of the tested mats were within the range 329–2589 mm/s. The 100% B mats showed the lowest air permeability. As the wool content in the mats increased, the air permeability gradually increased until it reached the value of 2539 mm/s for the 100% W mat.

### 3.2. Sound Absorption Coefficient, α, and Noise Reduction Coefficient, NRC

The relationships between the sound absorption coefficient, α, and the frequency of the sound wave for the analysed samples are shown in Figure 2. Based on the values of the sound absorption coefficient, α, the values of the noise reduction coefficient, NRC, were calculated, which are presented in Figure 3.

The values of the sound absorption coefficient, α, were in the range of 0.06–0.88 for the 100% B mat, for the mat made with 25% wool, α was 0.09–0.90, for the mat with 50% wool, α was 0.07–0.91, for the mat with 75% wool, α was 0.08–0.88 and for the 100% W mat, α was 0.07–0.84.

The highest value of the NRC coefficient (0.34) was obtained for a mat made from 25% wool fibres. The lowest value of NRC (0.28) was found for the 100% W mat. The noise reduction coefficient, NRC, decreased by 6% when the sheep wool content of the sample increased.

### 3.3. Thermal Resistance, R_ct_, and Thermal Conductivity, λ, Coefficients

The results of the thermal resistance, R_ct_, and heat conduction, λ, measurements of mats made of bast fibres, wool and bast fibres mixed with wool are presented in Table 2.

The 100% B mat showed the lowest thermal resistance (0.1365 m^2^ K/W), while the mat made with 75% wool showed the highest thermal resistance (0.2443 m^2^ K/W), the increase in this value was 79%. An increase in the wool content in the mats caused an increase in their thermal resistance.

The values of the thermal conductivity coefficients, λ, of the tested mats were in the range of 0.034–0.057 W/(m K), with the 100% B mat having the highest value of the λ coefficient, and the 100% W mat the lowest. With an increase in wool content in the tested mats, a reduction in thermal conductivity coefficient was observed. The value of the thermal conductivity of the 100% W mat decreased by 40% compared to the value obtained for the 100% B mat.

## 4. Discussion

The surface density of mats with higher than 25% wool content in the fibre blends was lower than that of mats with a predominant proportion of bast fibres, except for 75% W/25% B. This is due to the difference in fibre density. The wool fibre density of 1.30 g/cm^3^ was lower than that of bast fibres, which was 1.48–1.50 g/cm^3^; hence, 100% woollen products will have a lower surface density than bast fibre products, assuming that the structure of the compared products is similar. The exception to this rule is the surface density of the mat with a raw material composition of 75% W/25% B, which could be explained by the uneven thickness of the fleece layers that make up the final form of the mat. The fleece formed in the carding process has a delicate structure with visible thinning. From a technological point of view, it is impossible to achieve uniformity in the thickness of the individual layers of the overlapping fleece; therefore, 18 overlapping layers were used to minimise the impact of a single fleece on the thickness formation and surface density of the finished final mat. The standard deviation was determined for the results of the mat surface density test, which indicates the scatter of the results around the mean value. In the production process, the fleece is formed mechanically using a carding machine equipped with suitably developed loosening and fiberizing elements. There is a probability of uneven arrangement of fibres in the fleece, which can be reflected in the differentiation in the surface density. Nevertheless, no statistical significance was demonstrated between the values of mats with different raw material compositions, except for the mat made of 100% bast fibres. However, one should bear in mind that the produced mats are intended for technical purposes, where their thicknesses can be adjusted depending on need, e.g., by layering the mats on top of each other.

As the wool content of the mats increased, the air permeability gradually increased until it reached a value of 2539 mm/s for the 100% W mat. The 100% B mat showed the lowest air permeability, i.e., reaching a value of 329 mm/s. The air permeability of textiles depends on the structure of these materials, mainly their thickness, as well as the density of threads/fibres from which these materials are made and the weave/way of their arrangement. These factors influence the size of the spaces between the fibres, which determines the ability of the materials to be air permeable. Due to their elasticity, wool fibres create loose, fluffy structures with a large number of spaces between the fibres; therefore, wool mats are characterised by greater air permeability. On the other hand, the bast fibres, devoid of elasticity, form a compact structure where the fibres are densely arranged one on top of the other, thus limiting the presence of free spaces between the fibres, and thus limiting air permeability.

When analysing the results of the sound absorption tests at a low frequency sound wave, it can be concluded that for all mats, the α coefficient was low, with values from 0.06 to 0.16.

Above 500 Hz, the α coefficient increased with increasing frequency, and at 1000 Hz it reached a value of 0.37 for the mat containing 25% W and 75% B. At this frequency, the lowest value of the α coefficient (0.29) was achieved by the 100% W mat. For all variants of mats tested in the frequency range from 1000 to 1250 Hz, the sound absorption coefficient decreased. Above 1250 Hz, the α coefficient increased again for all mats and at the frequency of 2000 Hz, its highest values of 0.78 was recorded for the 100% B mat and of 0.77 for the mat with 25% sheep wool. At a frequency of 2500 Hz, a decrease in the α value for 100% B and 100% W mats was observed, while the mats made with the blends of wool and bast fibres resulted in their highest values for the α coefficient.

At the highest frequency at which the measurements were made (3150 Hz), the sound absorption coefficient reached its maximum values for all variants of mats. The blended mats containing the wool and bast fibres reached the highest values at the above-mentioned frequency. The mats made with 25% and 50% wool addition reached a value of α = 0.91, which means that more than 90% of the acoustic wave energy was absorbed by the mats, and less than 10% of the acoustic waves were reflected. The mat with 75% wool addition reached a value of α = 0.88, which means that 88% of the acoustic wave energy was absorbed by the mats and 12% of the acoustic waves were reflected.

Comparing the obtained results of the tests performed at a frequency of 2000 Hz of the mats with 100% B with a thickness of 7 mm and mats containing 25% and 50% wool with a thickness of 10 mm, for which the α values ware in the range (0.76–0.78), it can be concluded that these mats absorbed more sound than those used commercially and that are listed on internet sources [21,22], i.e., felts with a thickness of 5 mm and 25 mm, porous fibreboard with a thickness of 12.5 mm, carpet with a thickness of 6 mm and curtains with <0.2 kg/m^2^.

The values of the sound absorption coefficients, α, at 3150 Hz frequency were compared with the values of other materials determined at a frequency of 4000 Hz, due to the lack of relevant data in the literature. In this case, the values of the α coefficient for mats with 25% and 50% wool content were estimated at the level of 0.90 and 0.91. These mats achieved α values higher than most materials/products used commercially [21,22], except for mineral wool with d = 5 cm and typical glass wool acoustic wall and ceiling panels, which absorb sound completely (α = 1.00). However, it should be emphasised that these are not materials based on plant and animal natural fibres. According to the literature [17], the sound absorption coefficient is influenced, among others, by material thickness and air permeability. Comparing the results of mat tests in terms of the sound absorption coefficient, α, for a range of different frequencies (Figure 2) with the available literature data [21,22], it can be generally concluded that all variants of the mats developed in this study showed low sound absorption at low frequencies of 250–500 Hz. A significant improvement in the sound absorption of the mats was found at medium and high frequencies.

The developed mats containing 25% and 50% wool fibres were characterised by a greater thickness than the 100% B and 100% W mats, and a higher air permeability compared to the 100% B mat. Nevertheless, 100% W mats were characterised by the lowest sound absorption coefficient, α, for all tested frequencies, as well as the lowest noise reduction coefficient (NRC) in comparison to mats with different raw material compositions. The results of testing the ability of mats to absorb sounds showed that all variants of mats were characterised by a good ability to absorb sounds in the medium and high frequency range. The sound absorption capacity of the mat made from the 25% W/75% B mixture was greater than that of the mat made of 100% B. In this case, the addition of sheep wool in the blend caused an increase in the value of the noise reduction coefficient, NRC. The mat with 50% W/50% B reached a value for the NRC coefficient which was on a par with the 100% B mat. Further increasing the wool content in the mats caused a reduction in the sound absorption capacity of 6% for the 75% W/25% B and for the 100% W mats.

The statistical analysis of the results of the NRC coefficient (Kruskal–Wallis test), tested at the significance level of 0.05, showed that the type of non-woven fabric significantly influenced the value of the NRC (*p* = 0.0002). In order to check which samples were statistically significantly different from each other, a post hoc test was carried out, which showed statistically significant differences between the nonwovens: 25% W/75% B and 75% W/25% B, 25% W/75% B and 100% W and 50% W/50% B and 100% W.

When analysing the results of noise reduction factors, it should be stated that mats with more than 25% wool content were characterised by a lower noise reduction ability than mats with a predominance of bast fibres in the mixture. Due to their surface structure, wool fibres had a greater ability to reflect acoustic waves than bast fibres.

Comparing the obtained test results of the noise reduction coefficient, NRC, (Figure 3) to the literature data [18,19,23,24], it can be concluded that the NRC values of mats made from sheep wool and bast fibres fall in the range of values corresponding to various materials used in construction and interior spaces. Materials such as rock wool (9 cm) (NRC = 0.90–0.95), mineral wool (5 cm) (NRC = 0.50–0.75) and a professional acoustic panel (NRC = 0.75–1.0), have significantly higher NRC coefficient values than those indicated in this study. It should be noted, however, that the NRC factor depends on the thickness of the material and the produced mats based on wool and bast fibres had a relatively small thickness compared to the above-mentioned materials; therefore, it should be assumed that increasing the thickness of the mats would significantly affect the values of the NRC coefficient. Comparing the obtained NRC coefficients with, e.g., wood, glass, carpet placed on the wall, concrete, brick or plasterboard (for which the thicknesses are not given, but it should be assumed that they are thicker than the produced mats), it can be seen that the mats have higher NRC coefficient values and thus are more sound-absorbing.

Typically used insulation materials have a λ coefficient of 0.036–0.045 W/(m K). Materials with λ ≤ 0.035 W/(m K) are considered to be good heat insulators. The materials that achieve a thermal conductivity value of 0.014 W/(m K) are considered as the best insulators. According to this interpretation of the λ coefficient, it should be stated that 100% W mats and wool/bast fibres mixed with up to 50% of bast fibres are good insulating materials. A very good heat insulator is air, for which λ is 0.02 W/(m K). Hence, fluffy materials with numerous large spaces with trapped air between the fibres will be characterised by a higher thermal resistance compared to materials with densely packed fibres. Therefore, mats with a high wool content are characterised by the best thermal insulation properties in comparison to other tested mats.

The obtained values of the thermal resistance, R, of a mat of thickness d with respect to the thermal conductivity, λ, are consistent with the formula: R = d/λ, which is confirmed by the values presented in Table 2. The 100% B mat has the highest coefficient, λ = 0.057 W/(m K), and shows the lowest thermal resistance, R_ct_ = 0.1365 m^2^ K/W, at the same time. The same relationship occurred in all analysed cases. As the value of the coefficient λ decreased, the thermal resistance, R_ct_, increased. The values of the thermal resistance, R_ct_, for 100% W mats and mats with 75% wool were similar.

The results of statistical analysis of the λ coefficient (Kruskal–Wallis test), tested at the significance level of 0.05, showed that the raw material composition of nonwoven fabric significantly influenced the λ level (*p* = 0.0002). In order to check which samples were statistically significantly different from each other, a post hoc test was carried out, which showed statistically significant differences between the nonwovens: 100% B and 75% W/25% B, 100% B and 100% W and 25% W/75% B and 100% W.

By comparing the obtained results of the thermal conductivity coefficients, λ, of the developed mats, the values of which are in the range of 0.034–0.046 W/m K, with the literature data [2,18,43], e.g., with polystyrene, which has a λ value in the range 0.031–0.045 W/(m K) or mineral wool, for which λ is in the range 0.033–0.045 W/(m K), it can be concluded that the developed wool–linen–hemp mats can replace the currently used non-environmentally friendly insulation materials.

Future research will concern determining the sound absorption coefficient of mats in a reverberation chamber according to EN ISO 354 and EN ISO 11654.

## 5. Conclusions

Domestic natural raw materials in the form of sheep wool from mountain sheep and waste bast fibres produced at initial and proper carding stages in the process of spinning linen or hemp strings were used in this study. Both types of fibre are often treated as by-products of animal husbandry and bast fibre processing.

Analysing the properties of all developed mats with different raw material compositions, e.g., 100% W, 25% W/75% B, 50% W/50% B, 75% W/25% B and 100% B, it has been demonstrated that mats with a higher content of bast fibres showed better sound absorption properties, while mats with a higher content of wool were characterised by better thermal insulation.

The results of sound absorption tests showed that all variants of mats were characterised by a good ability to absorb sounds in the medium and high frequency range, while they had a lower ability to absorb sounds in the low frequency range.

The study confirmed the influence of the composition of the mats on the parameters of thermal insulation, i.e., thermal resistance and the thermal conductivity coefficient. Mats with a higher proportion of wool were characterised by higher thermal insulation compared to bast fibres mats. This is due to the fact that wool fibres are characterised by a lower thermal conductivity compared to bast fibres.

By analysing the values of the parameters characterizing the barrier properties in terms of both thermal insulation and sound absorption capacity tested in a specific frequency range and for specific raw material compositions of mats, it was found that a mat with a raw material composition of 50% W/50% B met both barrier criteria, i.e., it reached the values of thermal resistance and thermal conductivity coefficient to qualify it as a good heat insulator. In addition, it is a material that absorbs sounds and reduces noise to the levels of currently used soundproofing materials.

The tests conducted on mats allowed for an indication of their application character. Two types of natural waste material in the form of sheep wool and flax and hemp fibres show synergy with each other if they are used simultaneously as co-components in the mat’s composition. Synergy was observed in the area of sound absorption, where an improvement in the sound absorption of mats was achieved by adding bast fibres to the wool. It has been shown that both types of waste materials can be a valuable raw material used in the production of acoustic insulation materials. They can be applied as materials used in building partitions, or constitute the basis for acoustic panels, replacing polystyrene, mineral wool, glass wool and polyurethane foams commonly used in this field.

When considering the use of the results of this research in the technologies of thermal insulation and soundproofing of rooms with the use of sheep wool of local sheep breeds and waste bast fibres, it is necessary to take into account the need to use measures to prevent the growth of microorganisms and insects.

## Figures and Tables

**Figure 1 materials-16-00459-f001:**
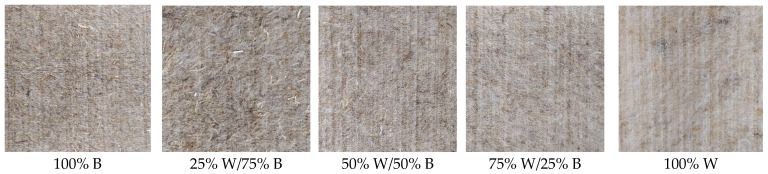
Photographs of the manufactured mats.

**Figure 2 materials-16-00459-f002:**
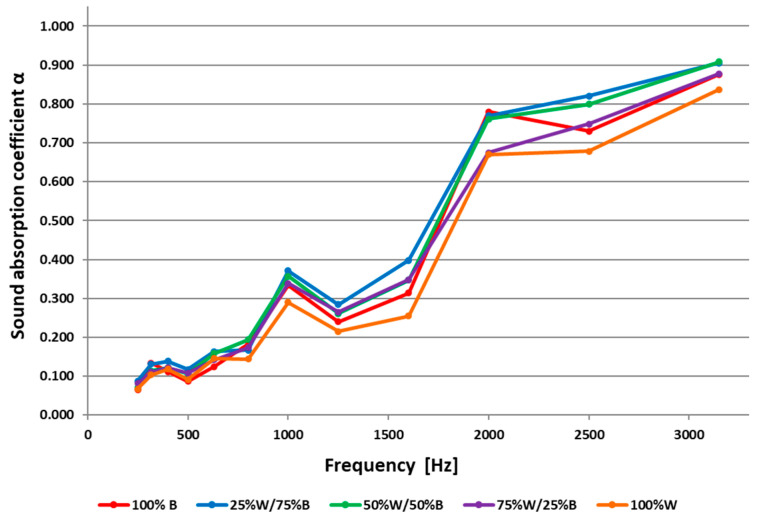
Sound absorption coefficient, α.

**Figure 3 materials-16-00459-f003:**
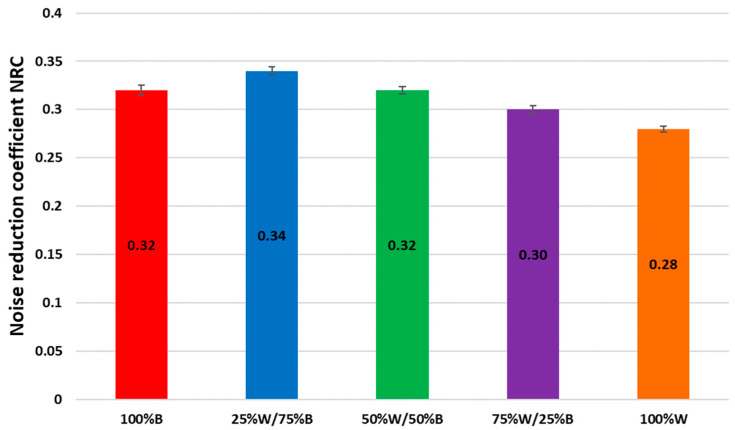
Noise reduction coefficient, NRC.

**Table 1 materials-16-00459-t001:** Basic parameters of mats produced from mountain sheep wool and flax and hemp waste.

Sample	Thickness			Surface Density			Air Permeability		
	AV mm	SD mm	CV %	AV g/m^2^	SD g/m^2^	CV %	AV mm/s	SD mm/s	CV %
100% B	6.8	0.2	3.6	1279	95	7	329	80	24
25% W/75% B	10.2	0.6	6.1	1491	206	14	620	137	22
50% W/50% B	10.5	0.8	8.0	1036	23	2	882	177	20
75% W/25% B	10.6	0.5	5.0	1330	126	9	1289	168	13
100% W	8.6	0.6	6.5	986	81	8	2539	390	15

**Table 2 materials-16-00459-t002:** Thermal resistance, R_ct_, and thermal conductivity, λ, coefficients.

Sample	Thermal Resistance R_ct_			Thermal Conductivity λ		
	AV m^2^ K/W	SD m^2^ K/W	CV %	AV W/mK	SD W/mK	CV %
100% B	0.1365	0.0031	2.2639	0.057	0.002	3.993
25% W/75% B	0.1967	0.0025	1.2749	0.046	0.002	5.265
50% W/50% B	0.2194	0.0054	2.4474	0.041	0.002	5.890
75% W/25% B	0.2443	0.0024	0.9752	0.037	0.002	6.209
100% W	0.2407	0.0037	1.5624	0.034	0.002	7.044

## Data Availability

Not applicable.

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
