# Peer review of "Effect of Blend Composition on Barrier Properties of Insulating Mats Produced from Local Wool and Waste Bast Fibres"

_materials, 2023, doi:10.3390/ma16010459_

Round 1

Reviewer 1 Report

This paper investigated Effect of blend composition on barrier properties of insulating mats produced from local wool and waste bast fibers. The most worthwhile contributions in this paper are the descriptions of novel wool and waste bast fibers composite materials made with thermal, and mechanical properties materials.

1.      In the introduction part, many details of the work have been discussed. Some paragraphs do not need to be reported due to the clarity of the acoustic content. Please summarize.

2.      An updated and complete literature recent review should be conducted as many researchers recently look at the sound absorption of natural fibers. In fact, many studies have looked at palm, kenaf, sugarcane fibers ad are not cited in this paper. see for example the study

 "Sugarcane bagasse waste fibers as novel thermal insulation and sound-absorbing materials for application in sustainable buildings". Building and Environment. 2022 Jan 8:108753.

 "Mathematical and experimental investigation of sound absorption behavior of sustainable kenaf fiber at low frequency". International Journal of Environmental Science and Technology. 2021 Sep;18(9):2765-80.

“Samaei SE, Berardi U, Soltani P, Taban E. Experimental and modeling investigation of the acoustic behavior of sustainable kenaf/yucca composites”. Applied Acoustics. 2021 Dec 1;183:108332.

“Abdi DD, Monazzam M, Taban E, Putra A, Golbabaei F, Khadem M. Sound absorption performance of natural fiber composite from chrome shave and coffee silver skin. Applied Acoustics”. 2021 Nov 1;182:108264.

3.      Please provide more information about the sample preparation and impedance tube system:

4.      How did you shape the samples to fit the impedance tube?

5.      Add reference and characteristics of the XXX Impedance Tube Solutions.

6.      It is necessary to highlight the measurement error and compare the measured values to those obtained by other authors.
Explain better how the measuring tube for the material was made. absorbers use two tubes of different diameters depending on the maximum measurement frequency. How did they merge the two different measures, how did they merge them? Do the samples when installed in the pipe get damaged in the assembly? How do they maintain the reproducibility of the measurement?

7.      More comparisons with updated studies are required in the results and discussion section, particularly within the last paragraphs.

8.      The conclusion section is too long and contains details that should be included within the introduction or results and discussion section.

9.      Discussion: Please highlight your study's strengths and limitations

10.  Discussion: Suggest adding a paragraph on directions for future research and practice.

Reviewer 2 Report

1. Too many keywords.

2. It is recommended to simplify the introduction.

3. The material is selected from Polish wool, whether the experiment is repeatable.

4. There are too few characterization data, such as the two materials are different in moisture absorption, which will obviously affect the sound insulation and heat transfer effect of the products.

5. What do you mean by materials that are not made from plant and animal materials? Please use the correct word.

Round 2

Reviewer 2 Report

the manuscript can be published in present form.